# Comparing proficiency of obstetrics and gynaecology trainees with general surgery trainees using simulated laparoscopic tasks in Health Education England, North-West: a prospective observational study

Zaibun N Khan [iD],[1] Donna Shrestha [iD],[2] Abdulwarith Shugaba,[2]
Joel E Lambert [iD],[2] Justin Clark [iD],[3,4] Elizabeth Haslett [iD],[5] Karolina Afors,[6]
Theodoros M Bampouras [iD],[7] Christopher J Gaffney [iD],[2] Daren A Subar[8]

**Correspondence to**
Dr Christopher J Gaffney;
c.gaffney@lancaster.ac.uk

## ABSTRACT

**Background** Training programmes for obstetrics and gynaecology (O&G) and general surgery (GS) vary significantly, but both require proficiency in laparoscopic skills. We sought to determine performance in each specialty.

**Design** Prospective, observational study.

**Setting** Health Education England North-West, UK.

**Participants** 47 surgical trainees (24 O&G and 23 GS) were subdivided into four groups: 11 junior O&G, 13 senior O&G, 11 junior GS and 12 senior GS trainees.

**Objectives** Trainees were tested on four simulated laparoscopic tasks: laparoscopic camera navigation (LCN), hand–eye coordination (HEC), bimanual coordination (BMC) and suturing with intracorporeal knot tying (suturing).

**Results** O&G trainees completed LCN (p<0.001), HEC (p<0.001) and BMC (p<0.001) significantly slower than GS trainees. Furthermore, O&G found fewer number of targets in LCN (p=0.001) and dropped a greater number of pins than the GS trainees in BMC (p=0.04). In all three tasks, there were significant differences between O&G and GS trainees but no difference between the junior and senior groups within each specialty. Performance in suturing also varied by specialty; senior O&G trainees scored significantly lower than senior GS trainees (O&G 11.4±4.4 vs GS 16.8±2.1, p=0.03). Whilst suturing scores improved with seniority among O&G trainees, there was no difference between the junior and senior GS trainees (senior O&G 11.4±4.4 vs junior O&G 3.6±2.1, p=0.004).

**Discussion** GS trainees performed better than O&G trainees in core laparoscopic skills, and the structure of O&G training may require modification.

**Trial registration number** ClinicalTrials.gov Registry (NCT05116332).

## INTRODUCTION

The foundations of laparoscopic surgery were laid by gynaecologists and the first sterilisation procedure was performed laparoscopically

### STRENGTHS AND LIMITATIONS OF THIS STUDY

⇒ The study's prospective design, robust data collection techniques including duplicate and blinded outcome assessment, and use of validated tools allowed us to minimise bias.

⇒ The study reported effect sizes as well SDs and CIs to allow the reader to assess the magnitude of study findings.

⇒ The generalisability of the study can be enhanced if the study is repeated on a national or international scale.

⇒ Larger comparative cohorts can provide more precision around the estimates of skill and allow adjustment for potential prognostic factors.

in 1936.[1] Gynaecologists have led advancements in laparoscopy through innovation in laparoscopic instruments and educational tools such as the pelvic simulator trainer and Hasson's open technique for entry, which is widely used by general surgeons today.[1 2]

Obstetrics and gynaecology (O&G) and general surgery (GS) trainees are required to demonstrate competency in different procedures[3 4]; however, the core psychomotor skills required for laparoscopy are similar. Some of these skills include laparoscopic camera navigation (LCN), hand–eye coordination (HEC) and bimanual coordination (BMC). Surgical trainees should be proficient in these skills early in their training to enable development of more complex and specific laparoscopic procedural techniques.[5 6]

O&G training, lasting 7 years, consists of basic (ST1–ST2), intermediate (ST3–ST5) and advanced training (ST6–ST7). The

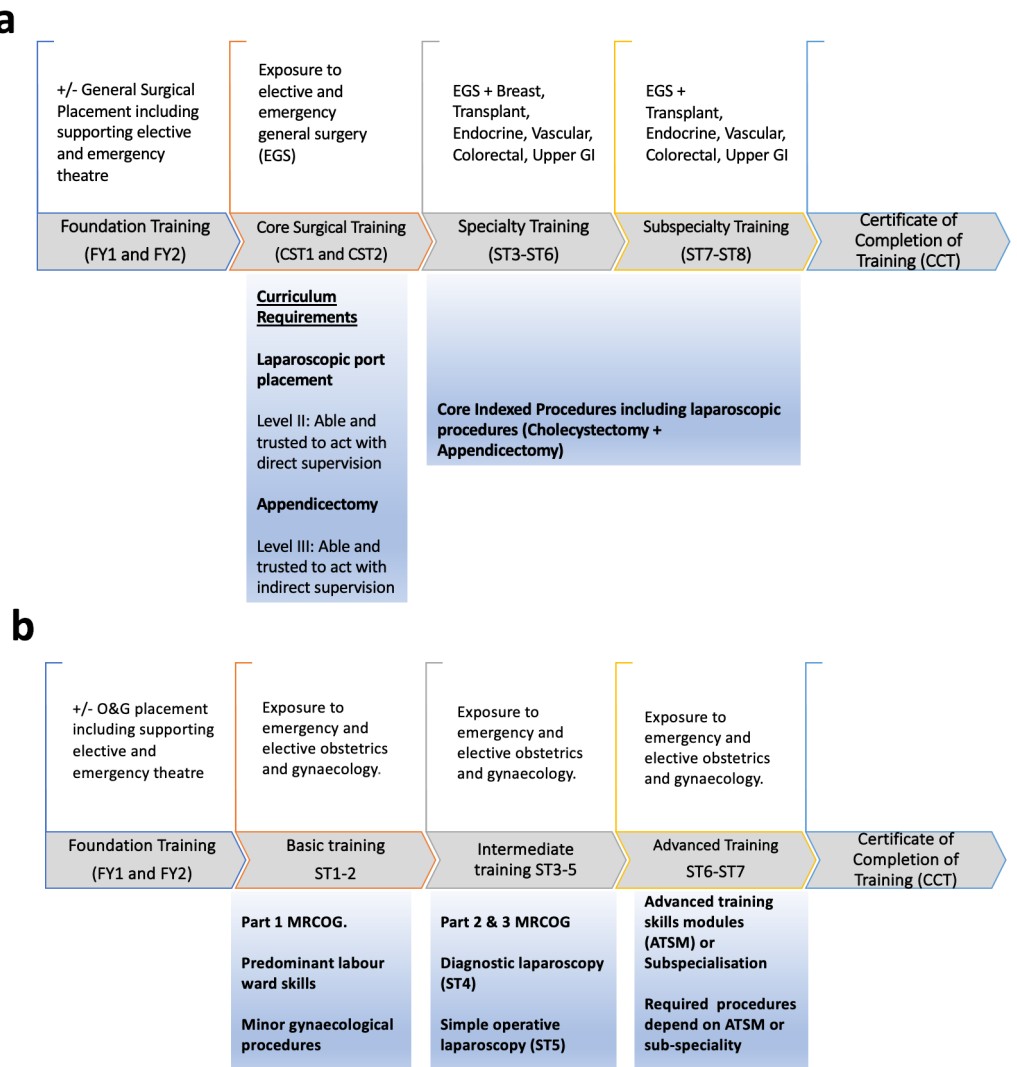

**Figure 1** Outline of the training pathways in GS (A) and O&G (B). Adapted from the Royal College of Obstetricians and Gynaecologists (RCOG)[3] and intercollegiate surgical curriculum programme.[4] GI, gastrointestinal; GS, general surgery; MRCOG, Member of the RCOG; O&G, obstetrics and gynaecology.

training covers both obstetrics and gynaecology, although there is a significant focus on acquiring obstetric competencies throughout the training.[7] Exposure to laparoscopic surgery is gained only through gynaecological practice. Trainees who wish to pursue gynaecological training can select Advanced Training Skills Modules or subspecialisation relevant to gynaecological surgery in the advance part of the programme.[8] In contrast, GS training is 8-years long, including 2 years of core surgical training (CST1–2) and 6 years of higher surgical training (ST3–ST8), where the final 2 years focuses on subspecialty training (figure 1).[4] GS trainees are required to be independent in laparoscopic appendicectomy by the end of CST2.[4] In contrast, O&G trainees are expected to perform 'minor operative laparoscopy' by the end of the fifth training year.[3] GS trainees, therefore, gain laparoscopic experience throughout their training programme, while O&G trainees receive most of their laparoscopic surgery exposure in the advanced part of the programme.[8 9] The content of each stage of laparoscopic training in O&G

and GS training is detailed in online supplemental tables 1 and 2.

Opportunities for theatre experience appear to be lacking in both specialties. In 2021, the Royal College of Obstetricians and Gynaecologists (RCOG) evaluated the training of 1415 trainees and found that less than half of the ST5 and ST6 trainees reported adequate opportunities to develop the required surgical skills relevant to their stage of training.[10] Similarly, among 155 GS applicants certifying for completion of training, only two-thirds had reached the required number of cases. However, nearly three-quarters of these trainees had met the requirements for key procedures in their field.[11]

Our study compared the proficiency in core laparoscopic psychomotor skills among junior (ST3–ST5 in both specialties) and senior trainees (ST6–ST7 in O&G; ST7–ST8 in GS) using a Karl Storz Szabo-Berci box trainer. We hypothesised that there is no difference in the performance of core laparoscopic skills between O&G and GS trainees at all training stages.

## METHODS

### Participants

Forty-seven trainees (24 O&G and 23 GS) from Health Education England North-West were invited to participate in this prospective observational study between September 2021 and April 2022. Trainees were allocated a study number, which was recognisable only to the two study investigators involved in the recruitment of trainees. To explore the effect of surgical experience, the trainees were subdivided by their training grades into four groups: junior O&G, senior O&G, junior GS and senior GS.

The 'junior' group consisted of trainees between ST3 and ST5, and the 'senior' group included trainees in the final 2 years of O&G and GS training programmes. For the senior O&G group, we selected trainees undertaking one of the advanced modules in 'advanced laparoscopy for the excision of benign disease', 'benign abdominal surgery-open and laparoscopic' and 'gynae-oncology'. This was to enable the selection of trainees in receipt of regular gynaecology theatre sessions and, therefore, comparable with GS seniors. Senior GS trainees with a specialist interest in breast surgery were excluded due to limited laparoscopic work within this subspecialty.

All participants provided written informed consent prior to participation. They completed a questionnaire collecting data on demographic details and factors relating to laparoscopic proficiency, such as the use of video games and laparoscopic simulators, attendance at courses involving laparoscopic surgery, training stage at first exposure to laparoscopic work and the typical frequency of attendance in theatre.

Following ethical approval, the study was registered at ClinicalTrials.gov (NCT05116332).

### Patient and public involvement

No patient involved.

### Procedures

All trainees were assessed by two faculty members/assessors in individual rooms to minimise external distractions. Assessors were not involved in the training of any study participants and trainees were able to discretely request a different assessor(s) if they knew the pre-assigned member or felt uncomfortable with them, without giving a reason. Trainees' specialty and training stage were concealed from the assessors to ensure anonymity of trainees and blinding of the assessors. Laparoscopic proficiency was measured by observing four standardised, simulated tasks using validated assessment tools.[5 12–14] All trainees received the same written and video instructions explaining the task before beginning any assessments.[15] All tasks were performed on a Karl Storz Szabo-Berci-Sackier laparoscopic trainer. The first three tasks assessed core laparoscopic psychomotor skills using the Laparoscopic Skills Training and Testing (LASTT) model.[13] The fourth task evaluated laparoscopic suturing and was assessed using the suturing and knot tying training and testing (SUTT-1) method by the European Academy of Gynaecological Surgery.[16] Trainees performed each task three times, except for the suturing task, which was completed once. The rationale behind restricting repetition to three iterations was to familiarise trainees with the task so that their optimal performance could be elicited without inducing a significant rehearsal effect.[17]

The same equipment was used throughout the testing period for all trainees. All assessors received standardised training modified from the 'Training the Trainers' of the Gynaecological Endoscopic Surgical Education and Assessment Programme. This consisted of an overview of all study tasks, instruments, scoring systems and specific details relating to set-up and delivery of all the study tasks. Tasks were performed in order of increasing technical difficulty as described below.

### Tasks

#### Task 1: LCN

This task assessed the trainees' ability to navigate a 30° 10 mm laparoscope to find 14 targets within the LASTT model.[12–14] The maximum time allowed was 300 s per iteration. A validation study on the LASTT model showed that the median time for task completion was 188 s for novices and ranged between 142 and 292 s.[12] O&G trainees use 30° telescope in hysteroscopic surgery and when using smaller laparoscopes. As the experience with using larger 30° laparoscopes may have been limited, we used the upper limit of the time range as the allocated time.

On the scoring sheet, the time taken to identify all 14 targets, or the last target identified within 300 s, was recorded. The task was considered successful when all 14 targets were identified in every iteration within the allocated time frame. The trainees' best time (of the three iterations) was used to assess the speed of task completion. To assess the trainees' ability to integrate speed with navigation skills, the ratio of the total number of targets found to the total time taken to complete the task was calculated.

#### Task 2: HEC

This task required the trainee to transfer six coloured cylinders to their respective coloured pins using forceps in their dominant hand and navigating a 0° laparoscope with their non-dominant hand.[12–14] Time permitted for this task was 180 s per iteration.[12 13]

Completion was determined when six cylinders were placed on their pins within the allocated time. The trainees' best time was used to calculate the speed of task completion. We recorded the total number of times a cylinder was dropped during each iteration. A sum of the three iterations gave a total number of drops. This was used as an indicator of precision of movement.

#### Task 3: BMC

This task assessed the trainees' ability to transfer six coloured pushpins between forceps in their dominant and non-dominant hands and place them in their coloured slots on the LASTT model.[12–14] The assessor navigated

**Table 1** Summary of laparoscopic surgical tasks using a box trainer and methods

| Task | Iterations | Time allocated | Data recorded | Outcome |
|---|---|---|---|---|
| 1. Laparoscopic camera navigation | 3 | 300 s | Time taken to find 14 targets. If exceeding 300 s, the last target found | Best time* Number of targets found |
| 2. Hand–eye coordination | 3 | 180 s | Time taken. Number of objects placed. Number of drops | Best time* Overall number of drops† |
| 3. Bimanual coordination | 3 | 180 s | Time taken. Number of objects placed. Number of drops | Best time* Overall number of drops† |
| 4. Suturing and intracorporeal knot placement (suturing) | 1 | 15 min | Time taken. Quality of sutures and knots | Median n° of sutures and knots. Total suturing scores |

*Shortest completion time out of three iterations.
†Sum of dropped objects across the three iterations.

the camera for the trainees based on their instructions. A maximum of 180 s was allowed per iteration and outcome measures were the same as for HEC.

### Task 4: laparoscopic suturing and intracorporeal knot placement (suturing)

A foam pad was used to assess suturing and knot placement using the SUTT-1 method.[16] All trainees were shown a 60-second video demonstration of laparoscopic suturing and intracorporeal knot tying to ensure that the instructions were standardised, and expectations were clearly understood.[18] Trainees were asked to place four interrupted sutures and perform four intracorporeal knots comprising of three throws. A maximum of 15 min was permitted for this task. The quality of suturing and knot tying was assessed by two experienced consultants (one O&G and one GS consultant; both with over 10 years of experience in laparoscopic suturing) after completion of the task using a validated SUTT scoring system.[16] The assessors were blinded to the trainee and each other's score. All components of the total suturing score, such as extent of trauma, were scored after thorough inspection of the foam pads. The suturing task was deemed complete if four horizontal sutures and four secure knots were secured within 15 min. The median number of sutures and knots inserted (out of four) and the total suturing scores were analysed.

A summary of the surgical tasks and their assessment is provided in table 1.

### Statistical analysis

The $\chi^2$ test was used to analyse demographic, training-related variables between specialties (table 2) and successful completion of all tasks. All continuous variables are reported as mean, SD and 95% CIs.

Normality was checked for tasks 1–3, including the LCN time and efficiency ratio, HEC time and precision score, and BMC time and precision score. As normality was only confirmed for BMC time, a robust analysis of variance (ANOVA)[19 20] was used to compare the junior and senior trainee groups within the two specialties. The Holm-Bonferroni post hoc test was carried out to locate the difference and adjust for multiple comparisons when a significant result was observed. Where trainee's surgical experience did not have a significant effect, robust independent t-tests were used to compare differences between O&G and GS. Effect sizes (ξ) were calculated for all significant comparisons, and 0.1 was considered small, 0.3 moderate and 0.5 large.[21] BMC time was analysed using ANOVA to compare junior and senior trainee groups within the two specialties and independent t-tests to assess differences between specialties. Holm-Bonferroni post hoc test was carried out to locate the difference and adjust for multiple comparisons when a significant result was observed.

In the suturing task, the numbers of sutures and knots were compared between the four groups using the Kruskal-Wallis test, with Holm-Bonferroni correction for multiple pairwise comparisons. These data are reported as median and IQR. Hedges' g was calculated for all significant comparisons with 0.2, 0.5 and 0.8 considered as small, moderate and large, respectively.[22] Agreement of total suturing scores between assessors was examined with Cronbach's α.[23] According to Bland and Altman, α=0.95 is desirable for clinical applications.[24] Total suturing scores were analysed using robust statistics as above. Statistical analysis was conducted in Jamovi V.2.3.18.0 (The Jamovi Project, https://www.jamovi.org), while collation and creation of figures were completed in GraphPad Prism V.9 (GraphPad Software, San Diego, California, USA). Statistical significance was set at p≤0.05 and the corrected values are presented.

## RESULTS

### Participant characteristics

Two trainees were excluded from the analysis as they did not meet the inclusion criteria (one senior O&G trainee) and had incomplete data (one senior GS trainee).

**Table 2** Laparoscopic training experience among O&G and GS trainees

| | O&G (n=23) | GS (n=22) | P value |
|---|---|---|---|
| Females | 15 (65%) | 5 (13%) | **0.004** |
| Males | 8 (35%) | 17 (77%) | |
| Juniors | 11 (48%) | 11 (50%) | 0.88 |
| Seniors | 12 (52%) | 11 (50%) | |
| Right handedness | 21 (91%) | 19 (86%) | 0.59 |
| Left/ambidextrous | 2 (9%) | 3 (14%) | |
| Played video games | 11 (48%) | 8 (36%) | 0.43 |
| Used pelvic simulator | 16 (70%) | 7 (32%) | **0.01** |
| Weekly | 2 (9%) | 0 (0%) | |
| Monthly | 3 (13%) | 2 (9%) | 0.32 |
| Less frequent | 18 (78%) | 20 (91%) | |
| Attended laparoscopic courses | 18 (78%) | 20 (91%) | 0.24 |
| Start of laparoscopic training | | | |
| Core training | 14 (61%) | 14 (67%)* | 0.69 |
| Registrar training | 9 (39%) | 7 (33%) | 0.69 |
| Elective theatre sessions | 64 (37%) | 110 (63%) | **<0.001** |
| Junior | 13 (20%) | 54 (49%) | **<0.001** |
| Senior | 51 (80%) | 56 (51%) | 0.30 |
| Emergency theatre sessions/month | 23 (19%) | 100 (81%) | **<0.001** |
| Junior | 10 (43%)† | 46 (46%) | **0.003** |
| Senior | 13 (57%) | 54 (54%) | **<0.001** |
| Type of exposure | | | |
| Juniors as operator | 4 (36%) | 9 (82%) | **0.03** |
| Juniors as assistant | 7 (64%) | 2 (18%) | **0.03** |
| Seniors as operator | 10 (83%) | 10 (91%) | 0.59 |
| Seniors as assistant | 2 (17%) | 1 (9%) | 0.59 |

Data are presented as frequencies (%). P values in bold indicate significant findings.
*One junior GS trainee did not answer.
†One junior O&G trainee did not answer.
GS, general surgery; O&G, obstetrics and gynaecology.

Twenty-three O&G trainees (mean±SD, age 34±4 years) and 22 GS trainees (34±5 years) were selected for data analysis. The O&G group consisted of 11 junior and 12 senior trainees and GS group consisted of 11 junior and 11 senior trainees. Both groups were not significantly different except for their gender. Most O&G trainees were female in contrast to GS, where the majority were male.

### Factors relating to proficiency in laparoscopic skills

Pretesting baseline questionnaires showed that a significantly larger number of O&G trainees used a simulator than GS trainees (O&G 16 (70%) vs GS 7 (32%), p=0.01).

However, the number of trainees using the simulator frequently, such as once a month, was similar between the two specialties (O&G 3 (13%) vs GS 2 (9%), p=0.32). O&G trainees reported attending significantly fewer elective and emergency laparoscopic theatre sessions (O&G 64 (37%) and 23 (19%) vs GS 110 (63%) and 100 (81%), p<0.001 for both comparisons). However, analysis by training grade showed that senior O&G and senior GS trainees attended a similar number of elective sessions (O&G 51 (80%) vs GS 56 (51%), p=0.30). Furthermore, junior O&G trainees were assigned to an assistant's role significantly more frequently than junior GS trainees (O&G 7 (64%) vs GS 2 (18%), p=0.05) (table 2).

### Successful completion of tasks

Overall, O&G and GS trainees had 69 and 66 attempts at each of the three core tasks, respectively. A smaller number of attempts were successfully completed by O&G trainees in comparison with GS trainees on all three tasks (LCN task: O&G 50 (72%) vs GS 64 (97%), p<0.001; HEC task: O&G 54 (78%) vs GS 64 (97%), p=0.001; BMC task: O&G 47 (68%) vs GS 62 (94%), p<0.001).

### Task completion times (speed)

There was a significant effect of specialty on completion times for LCN ($F_{(3,33)}=6.26$, p=0.005, HEC; $F_{(3,33)}=7.34$, p=0.002, BMC; $F_{(3,41)}=11.6$, p<0.001). Post hoc analyses showed significant differences between junior O&G and junior GS trainees only and no significant difference was found within the specialty groups (ie, between junior and senior trainees in either specialty). Between-group comparison showed that O&G specialty trainees were 73 s slower at completing LCN (O&G 166±56 (139 to 193) s vs GS 93±21 (83 to 103) s, t(21)=4.17, p<0.001, ξ=0.76). O&G trainees were also significantly slower at HEC (O&G 105±30 (90 to 119) s vs GS 67±13 (60 to 73) s, t(25.6)=3.98, p<0.001, ξ=0.66) and BMC task (O&G 139±32 (125 to 153) s vs GS 100±20 (92 to 109) s, t(43)=4.74, p<0.001, ξ=1.41) (figure 2A–C).

### Precision of movements (accuracy)

Specialty had a significant effect on the precision of movements in LCN ($F_{(3,33)}=8.23$, p=0.001) and BMC ($F_{(3,33)}=3.37$, p=0.04). However, no significant difference was found in the precision of movements in HEC ($F_{(3,33)}=0.96$, p=0.43). Post hoc analysis showed that greater trainee experience did not significantly affect precision outcomes on these tasks. Therefore, the data were analysed by overall specialty. Overall, in LCN, O&G trainees found fewer targets, in the given time, than GS trainees (O&G 0.09±0.04 (0.07 to 0.10) vs GS 0.16±0.03 (0.14 to 0.17), t(31.6)=5.27, p<0.001, ξ=0.82). In BMC, O&G trainees dropped a significantly greater number of pins than GS trainees (O&G 5.4±2.3 (4.3 to 6.6) vs GS 2.9±1.7 (2.1 to 3.8), t(32.8)=3.03, p=0.005, ξ=0.53). O&G and GS trainees both dropped similar number of cylinders during HEC task (O&G 3.5±2.7 (2.2 to 4.8) vs GS 2.3±1.6 (1.5 to 3.1), t(32.2)=1.23, p=0.22, ξ=0.27) (figure 2D–F).

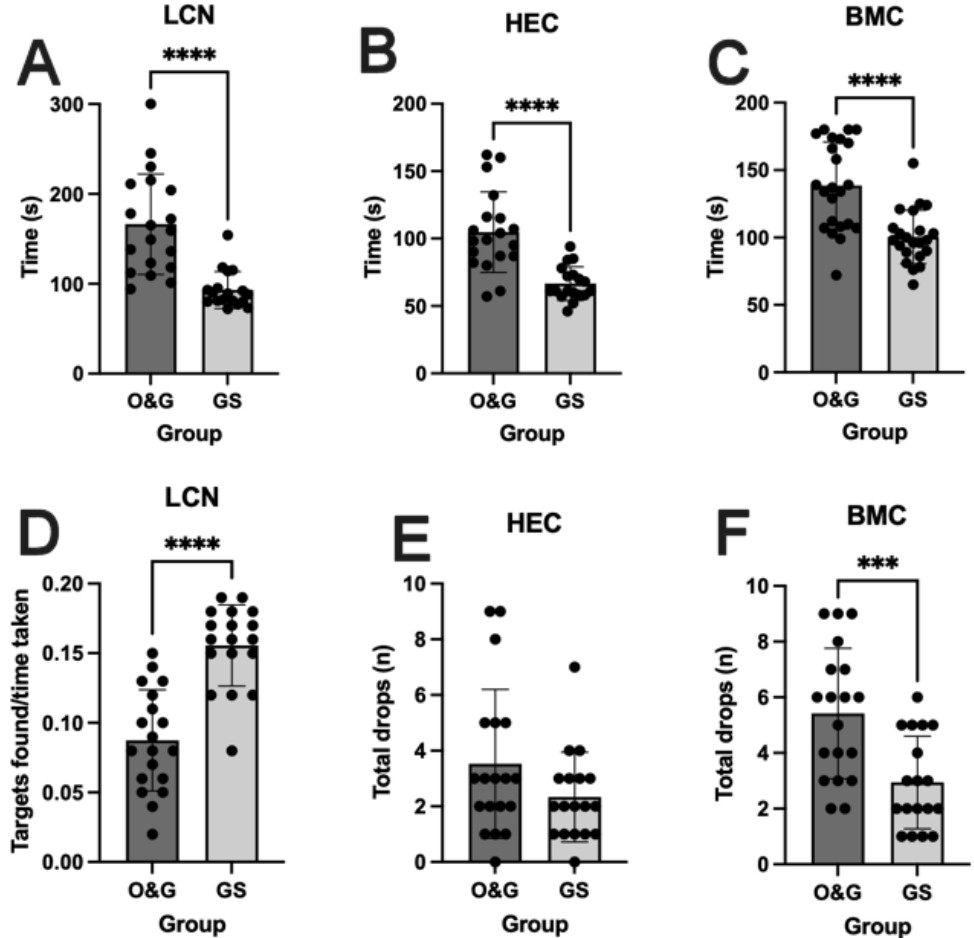

**Figure 2** Time taken to complete laparoscopic tasks (A–C) and laparoscopic precision of movements by specialty (D–F). Task completion time for LCN (A), HEC (B) and BMC (C). Trainees' ability to integrate camera navigation skills with speed (D), the number of drops in HEC (E) and the number of drops in BMC (F). Data are presented as mean±SD. BMC, bimanual coordination; GS, general surgery; HEC, hand–eye coordination; LCN, laparoscopic camera navigation; O&G, obstetrics and gynaecology. ***p<0.001; ****p<0.0001.

## Suturing
The inter-rater agreement of the assessors on the suturing task was very high (Cronbach's α=0.98 for O&G and 0.97 for GS). One O&G trainee (4.3%) and eight GS trainees (36%) completed this task in time (p=0.007).

## Number of inserted sutures and knots
Overall, O&G junior trainees were able to place fewer sutures and tie fewer intracorporeal knots than junior GS trainees (sutures: O&G 1 (1–1) vs GS 4 (3–4), p=0.005, Hedges' g=0.98; knots: O&G 0 (0–1) vs GS 2 (2–4), p=0.005, g=0.95). Senior O&G trainees tied significantly fewer knots than senior GS trainees (O&G 2.5 (1–3) vs GS 4 (3–4), p=0.03, g=0.51). However, senior trainees in O&G and GS groups placed similar number of sutures (O&G 3 (2–3) vs GS 4 (3–4), p=0.07, g=0.4).

## Total suturing scores
O&G trainees had a significantly lower total suturing score than the GS trainees (F(3,33)=36.3, p<0.001). Post hoc analysis showed that junior O&G trainees' total suturing score was significantly lower than junior GS trainees (O&G 3.6±2.1 (1.97 to 5.14) vs GS 14.9±4.4 (11.5 to 18.3), p<0.001) and senior O&G trainees also scored lower than senior GS trainees (O&G 11.4±4.4 (8.2 to 14.6) vs GS 16.8±2.1 (15.2 to 18.4), p=0.03). Senior O&G trainees had a significantly higher total suturing score than junior O&G trainees (senior O&G 11.4±4.4 (8.23 to 14.6) vs junior O&G 3.6±2.1 (1.97 to 5.14), p=0.004). Senior GS trainees, however, scored like their junior colleagues (senior GS 16.8±2.1 (15.2 to 18.4) vs junior GS 14.9±4.4 (11.5 to 18.3), p=0.35) (figure 3).

## DISCUSSION
### Principal findings
The acquisition of core laparoscopic skills depends on multiple factors including exposure to large volumes of laparoscopic procedures,[25] deliberate practice[26] and structured simulation programmes.[27] It is unknown whether the differing design of O&G and GS training leads to differential attainment of laparoscopic skills. Our study found that GS trainees performed better than O&G trainees in all tasks that measured core laparoscopic psychomotor skills. This may, in part, be due to

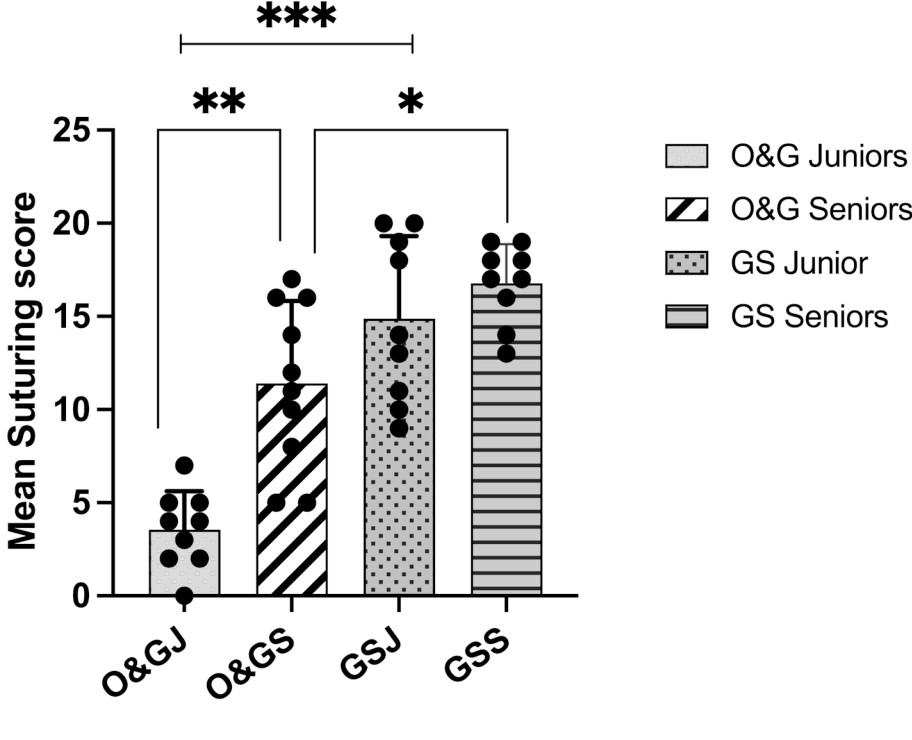

**Figure 3** Total suturing scores by trainee's experience within O&G and GS. Data are presented as mean±SD. GS, general surgery; GSJ, GS junior; GSS, GS senior; O&G, obstetrics and gynaecology; O&GJ, O&G junior; O&GS, O&G senior. *p<0.05; **p<0.01; ***p<0.001.

the discrepancy in the volume of laparoscopic practice between the two specialties. Our baseline questionnaire showed that the average GS trainees attended the operating theatre almost three times as often as the average O&G trainee and were more likely to perform as the main operator in contrast to O&G trainees.

Our study found that increased training experience had an impact on suturing and knot tying but not on the other three core laparoscopic tasks. This may be due to the simplicity of these core tasks. Surgical skills such as navigating a camera and retracting surgical tissue are usually learnt early in the training and reach a plateau phase rather quickly. It has been confirmed that participants rapidly reached their optimal performance on simple tasks such as HEC and that despite further training, no significant improvements were seen in performance.[5] Suturing, however, is regarded as a complex task and has been shown to improve with greater surgical experience.[28]

### Meaning of the study: possible explanations and implications for clinicians and policymakers

Most of the emergency work in O&G relates to obstetrics, and exposure to out-of-hours laparoscopic procedures is therefore limited.[29] Our study confirmed this. Overall, O&G trainees attended fewer laparoscopic theatre sessions and were less likely to be given the main operator's role than their GS counterparts. However, this difference was largely between the junior trainees only. Our baseline questionnaire showed that senior O&G trainees, in fact, attended a similar number of elective theatre sessions as the senior GS trainees and acted as the 'main operator' almost as frequently as the senior GS trainees. It appears that in O&G, theatre exposure and operative opportunities are concentrated in the latter part of the training. Psychological techniques have consistently shown that distributed practice is superior to concentrated practice and leads to the enhanced acquisition, consolidation and retention of surgical skills.[30 31] However, it remains unclear if the model of concentrated exposure in O&G may have contributed to the discrepancy in performance between the two specialties.

The RCOG expects all senior (advanced) trainees to be independent in laparoscopic salpingectomy (a procedure used for removing tubal ectopic pregnancy).[3] However, senior trainees' competency in salpingectomy has been shown to vary between 32% and 89%.[32 33] Based on feedback from O&G trainees, and documented benefits of distributed practice in learning new skills,[30 31] introducing salpingectomy earlier in the O&G curriculum might be helpful. It may encourage hospitals to give trainees more surgical exposure from an earlier stage, and trainees achieving competency in this simple procedure may find it easier to learn more complex skills such as laparoscopic suturing.[5]

A greater number of O&G trainees reported using a pelvic simulator; however, only a minority reported using it frequently. Surgical skills learnt on simulators can be

transferred to real patient surgery, but these benefits are mostly observed with repetitive practice and as part of a structured simulation programme.[34–36] The latter is promoted as a solution for bridging the gap between required operative skills and reduced training opportunities.[37 38] In this context, the American College of Obstetricians & Gynecologists has included a structured simulation programme, as part of board certification for practice in O&G.[39]

## Strengths and limitations of the study

To our knowledge, this is the first prospective study to examine trainees' laparoscopic skills in two surgical specialties who work in an anatomically similar environment. The training tools in this study were based on widely used and validated assessments,[12 13] and our inter-observer reliability for the suturing assessments was very high. The two assessors were not involved with the individual participants' training, and they were blinded to the trainee's specialty, experience and to each other's scores.

This study was localised to the North-West region of the UK and testing it on a national level would provide more precision around the estimates of skill and enhance external validity.

The effect of training grade was only apparent in the suturing and knot tying exercise. In the original study validating LASTT model, the novices were predominantly students with little or no operative experience and the experts were specialists with significant experience in advance surgical procedures. So, although the original study showed significant differences between novices and experts,[12] our junior group was more experienced than their novices. Therefore, it is possible that such differences were not large enough between our groups.

Simulation practice can facilitate the acquisition of new surgical skills if used systemically and comprehensively. Only a minority of the trainees undertook regular simulation and as such it is unlikely to have had a significant effect on the study tasks. Nonetheless, the type of simulation practice in this study has not been recorded, and this is a limitation.

The sample size may appear small for an observational study. Nonetheless, there are no previous studies available examining a similar aspect, and due to the difficulties in estimating the minimum difference considered important in this context, a priori sample size estimation was not possible. Consequently, along with the mean and SD values, we also included CIs and effect sizes to enable future meta-analysis as well as inform readers of the precision and magnitude of the results.

Finally, the male:female ratio between the specialty groups was considerably different, probably reflecting the relevant population in each specialty. Although evidence points to lack of differences between male and female surgeons,[40 41] future studies should aim to equate the participants based on sex, to alleviate any concerns around grouping male and female surgeons together.

## Unanswered questions and future research

The validity of evaluating core psychomotor skills in laparoscopic surgery needs to be assessed against actual performance in the operating theatre. Our work showed that trainees with limited experience found suturing (an actual surgical procedure) challenging but not the core psychomotor tasks. This implies that it is not just the mastery of core skills, but the cognitive and motor processes involved in applying these skills which may influence performance on actual surgical procedures. Therefore, future studies could look at cognitive and musculoskeletal stress among the two specialties and the seniority of its trainees.

**Author affiliations**
[1]Department of Gynaecology, Royal Lancaster Infirmary, Lancaster, UK
[2]Lancaster Medical School, Lancaster University, Lancaster, UK
[3]Department of Gynaecology, Birmingham Women's NHS Foundation Trust, Birmingham, UK
[4]Institute of Metabolism and Systems Research, University of Birmingham, Birmingham, UK
[5]North West School of Obstetrics & Gynaecology, Blackpool Victoria Hospital, Blackpool, UK
[6]Obstetrics & Gynaecology, Whittington Health NHS Trust, London, UK
[7]School of Sport and Exercise Sciences, Liverpool John Moores University, Liverpool, UK
[8]Department of General Surgery, East Lancashire Hospitals NHS Trust, Blackburn, UK

**Acknowledgements** We would like to thank Karl Storz, Medtronic, BSGE and BRIDGES society for their financial support with the study. We would also like to thank Brice Rodriguez, Kenn Ma, Dr Alison Sambrook, Kenn Lim, Mohamed Elsherbiny, Mohamed Abdelrahman, Christa Hammill, Aleyamma Abraham and Dr Nicholas Evans for their help with study protocol and conducting this study.

**Contributors** DaS and ZNK conceived and developed the research idea. ZNK, DoS, DaS, CG, TJC and EH designed and implemented the study protocol. ZNK, DoS, AS, JEL, TJC, EH, KA, TMB, CG and DaS conducted the study. TMB, CG and ZNK analysed the data. ZNK, DoS, AS, TJC, TMB, CG and DaS prepared the manuscript. All authors reviewed and approved the final manuscript. CG acts as guarantor of the work.

**Funding** This study was funded by grants from the British Society of Gynaecological Endoscopy, Karl Storz and Medtronic.

**Competing interests** None declared.

**Patient and public involvement** Patients and/or the public were not involved in the design, or conduct, or reporting, or dissemination plans of this research.

**Patient consent for publication** Not required.

**Ethics approval** This study involves human participants and was approved by the O&G and GS heads of schools from HEENW. Ethical approval was granted by the Faculty of Health and Medicine Research Ethics Committee at Lancaster University (FHMREC20033) and testing was conducted in accordance with the Declaration of Helsinki. Participants gave informed consent to participate in the study before taking part.

**Provenance and peer review** Not commissioned; externally peer reviewed.

**Data availability statement** Data are available upon reasonable request.

for any error and/or omissions arising from translation and adaptation or otherwise.

**ORCID iDs**
Zaibun N Khan http://orcid.org/0000-0002-9558-8964
Donna Shrestha http://orcid.org/0000-0001-8711-3515
Joel E Lambert http://orcid.org/0000-0001-5663-6284
Justin Clark http://orcid.org/0000-0002-5943-1062
Elizabeth Haslett http://orcid.org/0000-0003-1165-0415
Theodoros M Bampouras http://orcid.org/0000-0002-8991-4655
Christopher J Gaffney http://orcid.org/0000-0001-7990-2792

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
