## [Reviewer comments · BMJ Open]

ARTICLE DETAILS

TITLE (PROVISIONAL)	A prospective observational study comparing proficiency of obstetrics & gynaecology trainees with general surgical trainees using simulated laparoscopic tasks in Health Education England, North-West
AUTHORS	Khan, Zaibun N; Shrestha, Donna; Shugaba, Abdulwarith; Lambert, Joel E; Clark, T; Haslett, Elizabeth; Afors, Karolina; Bampouras, Theodoros M; Gaffney; Christopher; Subar, Daren

VERSION 1 – REVIEW

REVIEWER	Shen, Yang Southeast University, Obstetrics and Gynecology
REVIEW RETURNED	26-May-2023

GENERAL COMMENTS	This paper addresses an important and interesting problem—The effective differences of laparoscopy training in different specialties. A comparative approach between these two related surgical specialties may enable us to characterise the challenges associated with the acquisition of core laparoscopic skills in both O&G and GS trainees. But there are some suggestions for this article. 1. Please list the content of each stage of laparoscopic training in obstetrics and gynecology and surgical training in order to more intuitively understand the differences between the two laparoscopic training.2. Task Completion times(speed): no significant difference was found within the specialty groups, (i.e., between junior and senior trainees in either specialty). It is generally understood that advanced students perform laparoscopy faster than junior students. Can you explain this result?3. There are differences in laparoscopic operation ability between males and females, and the ratio of males and females in the two groups is significantly different. It is suggested to take gender as a stratification factor and conduct stratification data analysis, so that the results can be more reliable.
---

REVIEWER	Shore, Eliane St Michael's Hospital
REVIEW RETURNED	20-Jun-2023

GENERAL COMMENTS	Page 3 - discussion suggesting modification of training may be too large a conclusion to draw. Page 4 - is this how appendectomy is referred to "appendicectomy"
--

	Page 4 lines 43-47 don't belong in the introduction. They are an assumption. Page 5 - should explain you are using a box trainer (also in the introduction). I wasn't sure if you were talking about VR simulators and had to google the type of model (and I am a simulation expert). Page 6 - the first task chosen uses a 30 degree scope which the author's admit is not routinely used by gynecology. As a result, this may not be a reflective task of their surgical abilities. Is intracorporeal knot tying reflective of gynecologists surgical practice in England? It would be helpful to include some information on number of cases for competency especially when comparing case volumes of GS to O&G Page 11 - should not say "apparently" when discussing skills transfer from lab to OR. This has been proven in multiple studies. I don't understand why the FLS tasks were not used in this study, especially when the authors reference them in the discussion. It would have been interested to include urology which is more similar to obstetrics and gynecology than general surgery. The title of the paper is inflammatory. There is no explanation of what kills lab training the residents have. We do not know how many times the residents will have practiced the same tasks on the simulators and there is no discussion of this in the limitations.
--	---

VERSION 1 – AUTHOR RESPONSE

Reviewer 1

This paper addresses an important and interesting problem - The effective differences of laparoscopy training in different specialties. A comparative approach between these two related surgical specialties may enable us to characterise the challenges associated with the acquisition of core laparoscopic skills in both O&G and GS trainees.

Please list the content of each stage of laparoscopic training in obstetrics and gynecology and surgical training in order to more intuitively understand the differences between the two laparoscopic training.

Thank you for recognising the importance of this research area. We have included new supplementary tables (Table S1 and S2) cited in the introduction to provide further details on the training programmes of each specialty.

The following sentence has been added to the introduction:

The content of each stage of laparoscopic training in obstetrics and gynaecology and general surgery training is detailed in Table S1 and S2.

Table S1: Training matrix for O&G (adapted from RCOG). Required laparoscopic competencies are highlighted in bold. Competencies are signed off based on an entrustability scale* and as such no indicative numbers are included here.

	ST1	ST2	ST3	ST4	ST5	ST6	ST7
Curriculum	CiP progress appropriate to the relevant stage as per the CiP guides and entrustability levels.						
Examination	MRCOG Part 1			MRCOG Part 2 & Part 3			
At least 3 summative OSATS confirming competency by more than one assessor. At least one OSAT confirming competence should be supervised by a consultant.	Cervical smear	Caesarean section (Basic) Non rotational assisted vaginal delivery (Ventouse & forceps) Perineal repair Surgical management of miscarriage/Surgical termination of pregnancy Insertion of an Intrauterine system/intrauterine contraceptive device. Endometrial biopsy	Manual removal of placenta Transabdominal USS of early and late pregnancy	Hysteroscopy Diagnostic laparoscopy 3 rd degree perineal repair Vulval biopsy	Simple operative laparoscopy (laparoscopic sterilisation or simple adnexal surgery e.g. adhesiolysis/ ovarian drilling) Caesarean section (Intermediate) Rotational assisted vaginal delivery (any method)		Subspecialty specific competencies. Laparoscopic management of ectopic pregnancy Ovarian cystectomy (laparoscopic & open) Surgical management of post partum haemorrhage

*Entrustability scale: 1= observe only, 2= direct supervision, 3= indirect supervision, 4= act independently with support, 5= act independently.

Table S2: Summary of required procedures in GS training. Indicative case/operative numbers are given for the specialty training phase where both the numbers and entrustability scales are used for assessment.

	CT1 Phase 1	CT2 Phase 1	ST3 Phase 2	ST4 Phase 2	ST5 Phase 2	ST6 Phase 2	ST7 Phase 3	ST8 Phase 3
Examinations		MRCS Part A MRCS Part B						FRCS Part 1 FRCS Part 2
Operative Requirements Level 1 Has observed Level 2 Can do with assistance Level 3 Can do whole but may need assistance Level 4 Competent to do without assistance, including complications		Induction of pneumoperitoneum for laparoscopy with port placement (Level 2) Appendicectomy (Level 3) Open and close midline laparotomy incision (2) Inguinal hernia repair (Level 2) Primary abdominal wall hernia repair (Level 2)				Inguinal Hernia (level 4) [50 cases*] Cholecystectomy (level 3) [40 cases*] Segmental Colectomy (level 3) [15 cases*] Emergency Laparotomy [45* cases] Appendicectomy [60 cases*]		Emergency Laparotomy (Level 4) [100 cases*] Appendicectomy (Level 4) [80 cases*] Cholecystectomy [50 cases*] (level 4) Segmental colectomy [20 cases*] (level 4)
Other Operative Technical Skills		Chest drain insertion (Level 3) Needle biopsy including fine needle aspiration (Level 3) Rigid sigmoidoscopy (Level 3)						Indicative numbers and competencies for chosen specialty required. (Hepatopancreaticobiliary, Transplant, Endocrine, Colorectal, Oesophagogastric)

		Excision biopsy of benign skin or subcutaneous lesion (Level 4)						
--	--	---	--	--	--	--	--	--

2. Task Completion times (speed): no significant difference was found within the specialty groups, (i.e., between junior and senior trainees in either specialty). It is generally understood that advanced students perform laparoscopy faster than junior students. Can you explain this result?

The reviewer is correct to highlight this interesting finding. In our study there were 3 relatively simple (core) skills and 1 complex skill, the suturing task. We found that senior O&G trainees performed better than their junior counterparts on suturing and knot tying but this difference was not observed in the other 3 core tasks (LCN, HEC and BMC). Skills required to do the first 3 tasks are relatively simple and learnt early in the training, so the time in training to reach peak performance is quick (i.e. early in the training programme). Suturing, however, is a complex task that has a much steeper learning curve and is therefore more discriminatory.

We have added the following to our discussion:

Our study found that increased training experience had an impact on suturing and knot tying but not on the other three core laparoscopic tasks.... This may be the reason why senior trainees in this study differed from their junior colleagues only in the suturing performance.

The effect of training grade was only apparent in the suturing and knot tying exercise.

In the original study validating LASTT model, the novices were predominantly students with little or no operative experience and the experts were specialists with significant experience in advance surgical procedures. Furthermore, they showed that that with more practice the difference between the novices and experts decreased significantly. So, although the original study showed significant differences between novices and experts overall, our junior group was more experienced than their novices. Therefore, it is possible that such differences were not large enough between our groups.

3. There are differences in laparoscopic operation ability between males and females, and the ratio of males and females in the two groups is significantly different. It is suggested to take gender as a stratification factor and conduct stratification data analysis, so that the results can be more reliable.

We agree with the reviewer that the male: female ratio is substantially different between the two groups, reflecting the current gender representation in each specialty. The proposed subgroup

analysis would reduce the sample size further and create the risks identified before (and more recently, here (Riley et al., 2022), 6th day point). In other words, we would conclude on the difference between sexes (regardless of whether one way or the other) with inadequate precision to be confident in the result.

Adjusted analyses for potential prognostic variables would require a much larger sample to be reliable. We hope that our study can be replicated on a national / international level and adjusting for potential prognostic factors in statistical analysis would be possible and provide further insights.

Although there is some evidence that sex matters in laparoscopic skills e.g. (Schueneman et al., 1985), more recent and specific reports point towards lack of any differences (e.g. (Grantcharov et al., 2003), (Ali et al., 2015), (Madan et al., 2005), (Busshoff et al., 2022), (Kolozsvari et al., 2011)). It would, therefore, appear that currently there is not sufficient support to consider male and female surgeons differently; indeed, doing so in the absence of strong evidence could create the potential threat of falling victims to stereotyping (Myers et al., 2020)

Nonetheless, the point raised re the different male: female ratio between groups is a valid one. Therefore, we have added the following to the limitations to alert the reader to this fact:

Finally, the male: female ratio between the specialty groups was considerably different, probably reflecting the relevant population in each specialty. Although evidence points to lack of differences between male and female surgeons (e.g. (Ali et al., 2015; Kolozsvari et al., 2011)), future studies should aim to equate the participants based on sex, to alleviate any concerns around grouping male and female surgeons together.

We have also added a bullet in the “Strengths and limitations” section under the abstract related to this reviewer’s comment:

“Larger comparative cohorts can provide more precision around the estimates of skill and allow adjustment for potential prognostic factors.”

Reviewer 2

Page 3 - discussion suggesting modification of training may be too large a conclusion to draw.

We think this study provides some evidence that laparoscopic training should be introduced earlier in the O&G curriculum to give trainees longer exposure. Interestingly, the Royal College of Obstetricians & Gynaecologist (RCOG) is preparing to introduce the advanced training phase a year earlier so that

relevant competencies can be completed over three instead of the currently allocated two years. Our data provide some evidence to support this approach.

However, we recognise the concerns raised by the reviewer and agree our findings may have been overstated. We have modified the language and contents to temper our conclusions:

Based on feedback from O&G trainees, and documented benefits of distributed practice in learning new skills, introducing salpingectomy earlier in the O&G curriculum might be helpful. It may encourage hospitals to give trainees more surgical exposure from an earlier stage and trainees achieving competency in this simple procedure may find it easier to learn more complex skills such as laparoscopic suturing. ⁽¹⁾

Page 4 - is this how appendectomy is referred to "appendicectomy"

Thank you for pointing this out, appendicectomy is the British English spelling of appendectomy and we have used British English in our manuscript throughout as BMJ Open is a UK based journal.

Page 4 lines 43-47 don't belong in the introduction. They are an assumption.

Page 4 lines 43-47 are as follows: "The GS training program may be delivering better laparoscopic training than O&G. A comparative approach between these two related surgical specialties may enable us to characterise the challenges associated with the acquisition of core laparoscopic skills in both O&G and GS surgical trainees."

We recognise these two sentences could be considered speculative and have removed from the introduction.

Page 5 - should explain you are using a box trainer (also in the introduction). I wasn't sure if you were talking about VR simulators and had to google the type of model (and I am a simulation expert).

Thank you for this suggestion. We have modified the text as follows:

Our study compared the proficiency in core laparoscopic psychomotor skills ... using a Karl Storz Szabo Berci-Sackier box trainer.

And in the Method under *Procedures* as following.

All tasks were performed on a Karl Storz Szabo- Berci-Sackier box trainer.

Page 6 - the first task chosen uses a 30 degree scope which the author's admit is not routinely used by gynecology. As a result, this may not be a reflective task of their surgical abilities.

In O&G, 30 degree scopes are routinely used in hysteroscopic (vaginal) surgery and sometimes when using a smaller 5mm laparoscope. Some gynaecologists also use the angled (30°) 10mm scope, however, most frequently 10mm scopes in O&G are 0 degrees. The LASTT model and this task has been used for training purposes amongst O&G trainees with an angled scope. This task can only be achieved with a 30 degrees scope. We therefore concluded that it would be reasonable to use the angled scopes but allow a much longer time for completion (5 minutes compared with 3 minutes allowed in actual training on LASTT model).

We measured the proportion of trainees completing the task and found that significantly fewer O&G completed it than GS trainees. The performance on this task followed the same pattern as that on other tasks using 0 degrees laparoscopes, which suggests that the overall findings were consistent throughout the different tasks using both 0 degrees and 30 degrees laparoscopes. We agree with the reviewer that the use of a 30 degree scope can be explained more clearly.

The text in the methods have been amended as follows:

This task assessed the trainees' ability to navigate a 30° 10mm laparoscope ... 292 seconds. ⁽²⁾ O&G trainees use 30° telescope in hysteroscopic surgery and when using smaller laparoscopes. As the experience with using larger, 30° laparoscopes may have been limited, we used the upper limit of the time range as the allocated time.

We have also added the following text to the limitations:

Whilst gynaecologists use 30 degree scopes in hysteroscopic (vaginal) surgery, they are more commonly used in general surgery, and this may have affected the results in this task.

Is intracorporeal knot tying reflective of gynaecologists surgical practice in England?

This is an interesting point and the use of barbed sutures have transformed suturing practice amongst gynaecologists in UK. However, intracorporeal knot tying is still an essential requirement of some of the advanced training modules by the Royal college of obstetricians and Gynaecologist (Reference 8 in the manuscript) Indeed, the British Society for Gynaecological Endoscopy (BSGE) runs various

laparoscopic training programs and intracorporeal knot tying is almost always included in laparoscopic suturing.

It would be helpful to include some information on number of cases for competency especially when comparing case volumes of GS to O&G

Prior to the new curriculum in August 2021, the requirement for the number of cases that need to be logged at CCT in GS was 1600. However, since Aug 2021, the focus has shifted away from the requirement of an overall operative case number at CCT. Whilst indicative numbers are provided for index procedures (Table 1), competence can be achieved with fewer cases, if supported by procedural based assessments (PBAs) demonstrating performance at the level of a day-one consultant (PBA level 4). A 2013 study reviewing 58 GS trainee logbooks at CCT showed a mean of 607 (range 227-1384) cases logged at CCT. (Reference: Allum W, Hornby S, Khera G, Fitzgerald E, Griffiths G. General Surgery Logbook Survey. The Bulletin of the Royal College of Surgeons of England. 2013;95(4):1-6.)

O&G training does not require an indicative number of cases to show competence.

Procedure based competency in O&G is assessed by objective structured assessment of training (OSATs) using an entrustability scale. This scale is based on level of supervision; Level 1 is entrusted to observe, level 2 to act under direct supervision, level 3 as indirect supervision (supervisor available on site to provide direct supervision if needed), level 4 is entrusted to act independently with support (can call supervisor at home for support) and level 5 is entrusted to act independently.

Page 11 - should not say "apparently" when discussing skills transfer from lab to OR. This has been proven in multiple studies.

Thanks for this suggestion - we agree. This has been removed from the Discussion and it now reads:

Surgical skills learnt on simulators can be transferred to real patient surgery, but these benefits are mostly observed with repetitive practice and as part of a structured simulation program ⁽³⁻⁵⁾.

I don't understand why the FLS tasks were not used in this study, especially when the authors reference them in the discussion.

FLS tasks are primarily used in the US system. The Gynaecological Endoscopic Surgical Education and Assessment (GESEA) program is primarily used by UK trainees and is endorsed by the British Society for Gynaecological Endoscopy (BSGE). This is the European equivalent of FLS. The LASTT model and its tasks have been validated on trainees and consultants in Europe ⁽²⁾ and are therefore more representative of the UK system.

To remove confusion, we have removed the mention of FLS from the following sentence in the discussion:

In this context the American College of Obstetricians & Gynaecologist have included a structured simulation program, as part of board certification for practice in O&G. ⁽⁶⁾

It would have been interested to include urology which is more similar to obstetrics and gynecology than general surgery.

This is an interesting point. We agree that urologists and gynaecologists spend a lot of time operating in the pelvis. However, a UK GS trainee also spends a significant length of time learning to operate in the abdomino-pelvic area both electively and during emergency work.

The urology training structure is complex and the pool of trainees to recruit is limited. In UK, urology training can be entered in at least 2 ways, run through training or uncoupled by completing a core surgical training program first. Trainees entering the run through training will predominantly complete a urology themed curriculum whilst trainees from the uncoupled pathway will complete a general surgical curriculum. Thereby choosing urology trainees would have increased the number of training variables making head-to-head comparisons challenging. However, the reviewer suggests an interesting avenue for further research, which we are keen to pursue in future studies.

The title of the paper is inflammatory.

We don't intend to be inflammatory and have amended our title to a declarative statement that reflects our findings.

We have amended the title to be less inflammatory as follows:

Performance in simulated core laparoscopic skills is superior in general surgery than obstetrics and gynaecology trainees: a prospective observational study.

There is no explanation of what skills lab training the residents have. We do not know how many times the residents will have practiced the same tasks on the simulators and there is no discussion of this in the limitations.

Thank you for highlighting another great point. We enquired about use of pelvic simulators, but we agree that we did not record the type of simulation practice our trainees carried out. Our study found that although significantly more O&G than GS trainees had recently used a simulator, only a minority (O&G 2 (9%) vs GS 0 (0%)) used a simulator weekly. The benefits of simulation are observed with

regular and repetitive practice and neither of the two groups had done that. With infrequent simulator use, this factor was unlikely to have made a significant difference to the outcome of this study.

Nonetheless, we have added the following text to the limitations:

Simulation practice can facilitate the acquisition of new surgical skills if used systemically and comprehensively. Only a minority of the trainees undertook regular simulation and as such it is unlikely to have had a significant effect on the study tasks. Nonetheless, the type of simulation practice in this study has not been recorded, and this is a limitation.

Editor's comment-

Please remove the following sections as they are not requirements of BMJ Open:

What is already known on this topic

What this study adds

How this study might affect research, practice, or policy

These sections have now been removed from the manuscript.

Please include a 'Strengths and limitations of this study' section after the abstract. This section should contain up to five short bullet points, no longer than one sentence each, that relate specifically to the methods. The novelty, aims, results or expected impact of the study should not be summarised here. This will be published as a summary box after the abstract in the final published article.

Strengths and Limitations

- This is the first study to compare laparoscopic proficiency of trainees in obstetrics and gynaecology and general surgery using simulated tasks.
- The study's prospective design, robust data collection techniques, and validated tools allowed us to minimise bias.
- The study reported effect sizes as well standard deviations and confidence intervals to allow the reader to assess the magnitude of study findings.
- Absence of prior studies on the same subject prevented a priori sample size calculation.
- The generalisability of the study can be enhanced if the study is repeated on a national or international scale.
- Larger comparative cohorts can provide more precision around the estimates of skill and allow adjustment for potential prognostic factors.

References

- Ali, A., Subhi, Y., Ringsted, C., & Konge, L. (2015). Gender differences in the acquisition of surgical skills: a systematic review. *Surgical Endoscopy*, 29(11), 3065–3073. <https://doi.org/10.1007/s00464-015-4092-2>
- Busshoff, J., Datta, R. R., Bruns, T., Kleinert, R., Morgenstern, B., Pfister, D., Chiapponi, C., Fuchs, H. F., Thomas, M., Gietzelt, C., Hedergott, A., Möller, D., Hellmich, M., Bruns, C. J., Stippel, D. L., & Wahba, R. (2022). Gender benefit in laparoscopic surgical performance using a 3D-display system: data from a randomized cross-over trial. *Surgical Endoscopy*, 36(6), 4376–4385. <https://doi.org/10.1007/s00464-021-08785-4>
- Grantcharov, T. P., Bardram, L., Funch-Jensen, P., & Rosenberg, J. (2003). Impact of hand dominance, gender, and experience with computer games on performance in virtual reality laparoscopy. *Surgical Endoscopy And Other Interventional Techniques*, 17(7), 1082–1085. <https://doi.org/10.1007/s00464-002-9176-0>
- Kolozsvari, N. O., Andalib, A., Kaneva, P., Cao, J., Vassiliou, M. C., Fried, G. M., & Feldman, L. S. (2011). Sex is not everything: the role of gender in early performance of a fundamental laparoscopic skill. *Surgical Endoscopy*, 25(4), 1037–1042. <https://doi.org/10.1007/s00464-010-1311-8>
- Madan, A. K., Frantzides, C. T., Park, W. C., Tebbit, C. L., Kumari, N. V. A., & O’Leary, P. J. (2005). Predicting baseline laparoscopic surgery skills. *Surgical Endoscopy And Other Interventional Techniques*, 19(1), 101–104. <https://doi.org/10.1007/s00464-004-8123-7>
- Myers, S. P., Dasari, M., Brown, J. B., Lumpkin, S. T., Neal, M. D., Abebe, K. Z., Chaumont, N., Downs-Canner, S. M., Flanagan, M. R., Lee, K. K., & Rosengart, M. R. (2020). Effects of Gender Bias and Stereotypes in Surgical Training: A Randomized Clinical Trial. *JAMA Surgery*, 155(7), 552–560. <https://doi.org/10.1001/jamasurg.2020.1127>
- Riley, R. D., Cole, T. J., Deeks, J., Kirkham, J. J., Morris, J., Perera, R., Wade, A., & Collins, G. S. (2022). On the 12th Day of Christmas, a Statistician Sent to Me . . . *BMJ*, 379. <https://doi.org/10.1136/bmj-2022-072883>
- Schueneman, A. L., Pickleman, J., & Freeark, R. J. (1985). Age, gender, lateral dominance, and prediction of operative skill among general surgery residents. *Surgery*, 98 3, 506–515. <https://api.semanticscholar.org/CorpusID:24122791>

VERSION 2 – REVIEW

REVIEWER	Shen, Yang Southeast University, Obstetrics and Gynecology
REVIEW RETURNED	25-Aug-2023
GENERAL COMMENTS	As the authors state, this is the first study to compare the proficiency of Obstetrics & Gynecology and general surgery trainees in the use of laparoscopic simulated tasks. Although the study sample size was small and the selection of the population was limited, it cannot hide the highlights of this study. The study was prospective, used a replicated and blinded design, and the

	results are a very good guide for future exploration of stratified simulation training in laparoscopy.
--	--

VERSION 2 – AUTHOR RESPONSE

We would like to thank the additional reviewer for finding such merit in our work. We have amended the title as requested and removed point 1 of our strengths and limitations section.